# Structural bioinformatic study of six human olfactory receptors and their AlphaFold3 predicted water-soluble QTY variants and OR1A2 with an odorant octanoate and TAAR9 with spermidine

Finn Johnsson[1] ⬥, Taner Karagöl[2] ⬥, Alper Karagöl[2] ⬥ and Shuguang Zhang[3] ⬥

[1]St Paul's School, Lonsdale Road, London SW13 9JT, UK; [2]Istanbul Medical Faculty, Istanbul University, Istanbul, Turkey and [3]Laboratory of Molecular Architecture, Media Lab, Massachusetts Institute of Technology, 77 Massachusetts Avenue, Cambridge, MA 02139, USA

## Research Article

**Keywords:**
hydrophobic to hydrophilic conversion; membrane proteins; protein design; QTY code; water-soluble olfactory receptors

**Corresponding author:**
Shuguang Zhang;
Email: Shuguang@MIT.EDU

T.K. and A.K. contribute equally.

## Abstract

The molecular mechanism of olfaction, namely, how we smell with limited olfactory receptors to recognize exceedingly diverse and large numbers of scents remains unknown despite the recent advances in chemistry, chemical, structural, and molecular biology. Olfactory receptors are notoriously difficult to study because they are fully embedded in the cell membrane. After decades of efforts and significant funding, there are only three olfactory receptor structures known. To understand olfaction, we carried out the structural bioinformatic study of six human olfactory receptors including OR51E1, OR51E2, OR52cs, OR1A1, OR1A2, TAAR9, and their AlphaFold3 predicted water-soluble QTY variants with odorants. We applied the QTY code to replace leucine (L) with glutamine (Q), isoleucine (I) and valine (V) with threonine (T), and phenylalanine (F) with tyrosine (Y) only in the transmembrane helices. Therefore, these QTY variants become water-soluble. We also present the superimposed structures of native olfactory receptors and their water-soluble QTY variants. The superimposed structures show remarkable similarity with RMSDs between 0.441 and 1.275 Å despite significant changes to the protein sequence of the transmembrane domains (43.03%–50.31%). We also show the differences in hydrophobicity surfaces between the native olfactory receptors and their QTY variants. Furthermore, we also used AlphaFold3 and molecular dynamics to study the odorant octanoate with OR1A2 and spermidine with TAAR9. Our bioinformatics studies provide insight into the differences between the hydrophobic helices and hydrophilic helices, and will likely further stimulate designs of water-soluble integral transmembrane proteins and other aggregated proteins.

## Introduction

There are commonly known five basic senses including vision, auditory, touch, smell, and taste. We use these senses to interact with the external world. Each of these senses involves a specific class of biological membrane receptors in the body. These senses are essential for us to sense and interact with the external world. We have gained a great deal of knowledge of the molecular basis for vision, auditory, and touch, but we are still far from understanding the molecular basis of smell. Smell uses combinatorial limited olfactory receptors to recognize an exceedingly diverse and wide range of odorants and scents.

Human olfactory receptors (Buck and Axel, 1991) belong to a diverse family of G-protein-coupled receptors (GPCRs) (Firestein, 2001; Gaillard et al., 2004; García-Nafría and Tate, 2021) primarily involved in the detection of odorant molecules. These receptors are located on the olfactory sensory neurons in the olfactory epithelium of the nasal cavity. However, recent studies have revealed their ectopic expression in various non-olfactory tissues (Massberg and Hatt, 2018) including blood, breast, lung, intestine, skin, heart, prostate, liver, lungs, testis, and hair (Chéret et al., 2018) and overexpressed in some cancer cells (Chung et al., 2022). Studying olfactory receptors thus opens new avenues for frontier research including cancer research.

The olfactory receptors all have seven transmembrane alpha-helices (TM1–TM7) (García-Nafría and Tate, 2021). These helices form a cylindrical structure that traverses the cell membrane's lipid bilayer. The arrangement of these helices creates a binding pocket within the membrane where odorant molecules can interact with the receptor. The extracellular loops connecting these helices are also responsible for recognizing and binding odorant molecules. The intracellular loops and the C-terminal tail interact with G-proteins. The binding of an odorant molecule to the receptor activates the G-protein, which then triggers a signaling cascade involving

adenylate cyclase, which converts ATP to cyclic AMP (cAMP). The increase in cAMP levels leads to the opening of ion channels, resulting in neuronal depolarization and the transmission of the olfactory signal to the brain (Firestein, 2001; Gaillard et al., 2004; Glezer and Malnic, 2019; Kuroda et al., 2023).

The transmembrane domains of the olfactory receptors, like other integral membrane proteins, are composed of mostly hydrophobic amino acids that interact with the fatty acid chains of the membrane lipids, excluding water and rendering the transmembrane domains hydrophobic. For this reason, it is notoriously difficult to study membrane proteins as they require detergents to solubilize and stabilize these membrane proteins when they are removed from their membranes. This severely limits their application for medicinal and technological development (Vinothkumar and Henderson, 2010).

Olfactory receptor classes have been one of the most difficult membrane proteins to undertake structural studies because they are likely unstable without their odorants which often bind with low affinities. Among ~400 human olfactory receptors (Niimura et al., 2014), currently there are only one structure is available (Billesbølle et al., 2023), one engineered OR52cs with consensus protein sequences (Choi et al., 2023) and an amine odorant receptor TAAR9 receptor (Guo et al., 2023). Thus, alternative methods should be encouraged to study olfactory receptors.

We here selected six olfactory receptors to carry out structural bioinformatic studies.

OR51E1, also earlier known as PSGR (Prostate Specific G-Protein Coupled Receptor), is a member of the olfactory receptor family predominantly expressed in the prostate (Bax et al., 2018). It has been identified as a potential biomarker for prostate cancer because of its overexpression in malignant prostate tissues compared with benign ones (Weng et al., 2005, 2006). The activation of OR51E1 has been shown to suppress growth in human prostate cancer cells, therefore making it a possible alternative therapeutic target for prostate cancer (Massberg et al., 2016). Additionally, OR51E1 has been found to act as a tumor biomarker for lung carcinoids (LC) in somatostatin receptor-negative tumor patients (Giandomenico et al., 2013).

OR51E2, also known as PSGR2, is also a member of the olfactory receptor family predominantly expressed in the prostate. It is similarly overexpressed in malignant prostate cancer tissues compared with benign tissues (Weng et al., 2006). Activation of OR51E2 by specific ligands has been shown to evoke an intracellular calcium response and inhibit prostate cancer cell proliferation, suggesting its potential as a candidate for prostate cancer treatment (Neuhaus et al., 2009, Mermer et al., 2021).

OR1A1 is a member of the olfactory receptor family (Schmiedeberg et al., 2007). It has been detected that OR1A1 is significantly expressed on the surface of HepG2 liver cells. The activation of OR1A1 by the ligand (−)-carvone increases the cyclic adenosine monophosphate (cAMP) without changing the intracellular $Ca^{2+}$ concentration, thus inducing the protein kinase A (PKA) – cAMP response element-binding protein (CREB) – hairy and enhancer of split (HES)-1 signaling axis. In those cells where OR1A1 was activated by (−)-carvone, intracellular triglyceride levels were reduced. These results suggest that OR1A1 may modulate hepatic triglyceride metabolism (Wu et al., 2015).

OR1A2 is another olfactory receptor that is ectopically expressed in several tissues including the liver, blood, heart, and pancreas (Massberg and Hatt, 2018). OR1A2 is being explored for its huge therapeutic potential in the reduction of hepatocellular carcinoma progression. It has been found to be expressed in studies involving Huh7 cells, a monoterpene-activated hepatocellular carcinoma (HCC) cell line. Activation of OR1A2 by (S)-(−)-citronellal in these cells not only induces calcium signaling but also reduces cell proliferation (Massberg et al., 2015).

TAAR9 is an olfactory trace amine receptor belonging to the family of trace amine receptors. They are expressed in olfactory epithelium neurons. They detect diverse ethological signals including predators, decayed food, pheromones, and others (Gainetdinov et al., 2018). TAAR9 has been detected in breast cancer tissues. Recent studies show that there are correlations between the expression levels of TAAR9 and genes involved with neuroactive ligand signaling in abnormal tissue growth. This co-expression between genes in primary tumors and metastatic lesions suggests that TAAR9 may play a role in modulating breast cancer progression (Vaganova et al., 2023). Furthermore, in melanoma, deregulation of TAAR9 and other TAARs has been observed, indicating they may have significance in tumor progression (Vaganova et al., 2022).

OR52cs is a designed olfactory receptor combining the protein consensus sequences that represent 26 members of the human OR52 family (Ikegami et al., 2020). Its CryoEM structure has been elucidated (Choi et al., 2023). It is only one of three olfactory receptor structures so far among ~400 human olfactory receptors (Niimura et al., 2014).

In this structural bioinformatics study, we used the newly released AlphaFold3 (Abramson et al., 2024). AlphaFold3 is a deep-learning artificial intelligence model that uses a diffusion network to predict protein structures with incredible accuracy. Results that may have previously taken hours for AlphaFold2 were ready within minutes. AlphaFold3 is also capable predict complex interactions albeit not in the perfect state yet.

Google DeepMind released AlphaFold2 in July 2021 (Jumper et al., 2021; Jumper & Hassabis 2022; Varadi et al., 2022) and AlphaFold3 in May 2024 (Abramson et al., 2024). The Alpha-Fold2 and AlphaFold3 use artificial intelligence tools and deep learning to predict protein structures from their amino-acid sequences. They have revolutionized 3D protein structure predictions and AlphaFold3 is capable of predicting protein–molecular complexes. DeepMind, in partnership with EMBL-EBI, released the AlphaFold Protein Structure Database which contains over 214 million predicted protein structures (Varadi et al., 2024). In comparison to the ~224,000 experimentally determined structures available through RCSB-PDB, AlphaFold3 predictions have acknowledged limitations, and need to be validated through experimental analysis. Physical structural studies of the water-soluble QTY variants of membrane proteins are still needed to validate the AlphaFold3-predicted structures.

We previously applied the QTY (Glutamine, Threonine, Tyrosine) code to design several detergent-free chemokines and cytokine receptors, all of which retained structural thermal stability and native ligand-binding activities and enzymatic activities despite substantial changes to the transmembrane domain (Zhang et al., 2018; Hao et al., 2020; Tegler et al., 2020; Zhang and Egli, 2022; Li et al., 2024). These water-soluble variants were then used to elucidate the mechanism of native receptor-ligand interaction and their binding abilities despite significant truncation in several chemokine receptors (Qing et al., 2019; Qing et al., 2021, Qing et al., 2022). Using the online version of AlphaFold2, we predicted the QTY variant structures of 7 chemokine receptors and 1 olfactory receptor (Skuhersky et al., 2021), 14 glucose

transporters (Smorodina et al., 2022a) and 13 solute carrier transporters (Smorodina et al., 2022b), and 6 human ABC transporters (Pan et al., 2024), 8 human glutamate transporters (Karagöl et al., 2024a) and 7 human monoamine transporters (Karagöl et al., 2024b). We recently also showed that the QTY code also works very well for bacterial outer membrane beta-barrels (Sajeev-Sheeja et al., 2023), and for IgG monoclonal antibodies that are rich in beta-sheet structure (Li et al., 2023).

Recently, we have asked if the QTY code is applicable to other olfactory receptors. The olfactory receptors are all integral membrane proteins with seven transmembrane alpha-helices embedded in a lipid bilayer. Therefore, because of the hydrophobic properties of transmembrane domains, they are not water-soluble without the aid of detergents. We wanted to see if the QTY code could be utilized to design water-soluble variants of these olfactory receptor proteins.

Here we report the structural bioinformatic studies of three experimentally-determined olfactory receptors (OR51E2, OR52cs, and TAAR9), and three without experimentally-determined structures (OR1A1, OR1A2, OR51E1) and their AlphaFold3 predicted water-soluble QTY variants. We provide superpositions of the hydrophobic native transporters and their hydrophilic QTY variants. We also provide the comparative hydrophobicity molecular structures with their hydrophilic QTY variants. Furthermore, we provide the AlphaFold3 predicted and molecular modeled odorant binding studies of OR1A2 with its odorant octanoate and TAAR9 with its odorant spermidine.

## Results and discussion

### The QTY code

The QTY code is a straightforward tool devised to create water-soluble versions of membrane proteins, traditionally challenging because of their hydrophobic nature, thereby facilitating more effective research and drug development. When applying the code, the four hydrophobic amino acids, leucine (L), isoleucine (I)/valine (V), and phenylalanine (F), in the transmembrane domain, are pairwise replaced with the three polar and neutral amino acids glutamine (Q), threonine (T), and tyrosine (Y). Specifically, glutamine replaces leucine, threonine (T) replaces both isoleucine (I) and valine (V), and tyrosine replaces phenylalanine. This works because of the strong similarities of the electron density maps between Q versus L, T versus I&V, and Y versus T (Zhang et al., 2018; Zhang and Egli, 2022). Consequently, the hydrophobic amino acids in the transmembrane alpha-helical domains are replaced with hydrophilic amino acids upon applying the QTY code. Therefore, transforming the transmembrane and its properties from hydrophobic to water-soluble.

### Olfactory receptor protein sequence alignments and other characteristics

The protein sequences of the native olfactory receptors and those of their QTY variants were aligned (Table 1, Figure 1), revealing 20.98%–25.88% alterations in the overall amino acid composition, specifically substantial replacement in the transmembrane domains 43.03%–50.31%. Despite these changes, the isoelectric-focusing points (pI) remain similar, with anywhere from a 0- to 0.15-unit difference. These pI changes are inconsequential with respect to surface charges and unlikely to interfere with structures. At neutral PH, amino acids Q, T, and Y do not bear any charges; hence, they do not notably change a protein's pI after the QTY code has been applied. Instead of saturated carbon side chains, the Q, T, and Y amino acids have water-soluble side chains. The sidechain $-NH_2$ of glutamine (Q) can form four hydrogen bonds with water molecules (two as donors from $-NH_2$ and two as acceptors from the oxygen on $-C=O$), and the sidechain $-OH$ of threonine (T) and tyrosine (Y) can form three hydrogen bonds (two as acceptors from O and 1 as a donor from H). This explains why the molecular weights (MWs) of the QTY counterparts were slightly bigger than the native proteins, as Nitrogen (14 Da) and oxygen (16 Da) in the water-soluble side chains have a higher MW than carbon (12 Da).

**Table 1.** The protein characteristics of native olfactory receptors and their QTY variants

| Name | RMSD (Å) | pI | MW (kDa) | TM variation (%) | Overall variation (%) |
|---|---|---|---|---|---|
| OR51E2 (8F76) | — | 9.16 | 35.49 | — | — |
| OR51E2$^{QTY}$ | 1.275 Å | 9.04 | 35.88 | 46.62 | 21.88 |
| OR52cs (8HTI) | — | 9.44 | 35.29 | — | — |
| OR52cs$^{QTY}$ | 1.038 Å | 9.29 | 35.81 | 50.31 | 25.88 |
| TAAR9 (8IWE) | — | 6.22 | 39.01 | — | — |
| TAAR9$^{QTY}$ | 0.987 Å | 6.22 | 39.54 | 49.32 | 20.98 |
| OR51E1 | — | 8.71 | 35.40 | — | — |
| OR51E1$^{QTY}$ | 0.713 Å | 8.64 | 35.70 | 49.66 | 23.27 |
| OR1A1 | — | 8.96 | 34.56 | — | — |
| OR1A1$^{QTY}$ | 0.713 Å | 8.87 | 34.85 | 43.05 | 21.04 |
| OR1A2 | — | 8.81 | 34.34 | — | — |
| OR1A2$^{QTY}$ | 0.732 Å | 8.75 | 34.71 | 43.03 | 21.04 |

Abbreviations: Isoelectric focusing (pI), molecular weight (Mw), transmembrane (TM), not applicable (—), and residue mean-square distance (RMSD). The 5 olfactory receptors and TAAR9 are listed in the same order as Figure 1. RMSDs were calculated in the native cryo-EM determined structures and the corresponding residuals in the predicted QTY structures. The QTY amino acid substitutions in the transmembrane (TM) are significant between 43.03% and 50.31%, whereas the overall structural changes are between 20.98% and 25.88%.

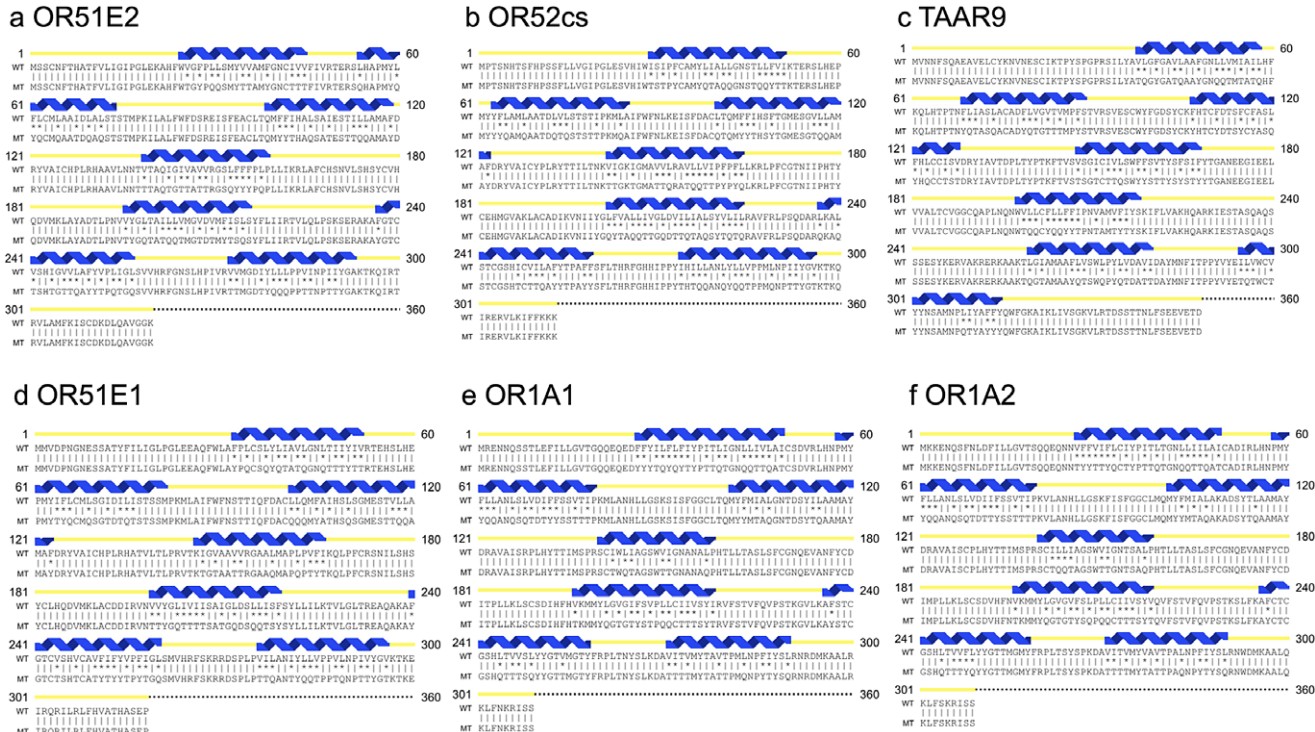

**Figure 1. Protein sequence alignments of six native olfactory receptor proteins with their water-soluble QTY variants**. The symbols | and * indicate whether amino acids are identical or different, respectively. Note the Q, T, and Y amino acids replacing L, V and I, and F, respectively. The alpha helices (blue) are shown above the protein sequences. The characteristics of natural and QTY variants listed are isoelectric focusing (pI), molecular weight (MW), total sequence variation 20.98%–25.88%, and transmembrane variation 43.03%–50.31%. The alignments are: a) OR51E2 versus OR51E2$^{QTY}$, b) OR52cs versus OR52cs$^{QTY}$, c) TAAR9 versus TAAR9$^{QTY}$, d) OR51E1 versus OR51E1$^{QTY}$, e) OR1A1 versus OR1A1$^{QTY}$, and f) OR1A2 versus OR1A2$^{QTY}$. Although there are significant QTY changes in the TM alpha helices (43.03%–50.31%), their changes in MW and pI are insignificant.

## Side-by-side display of the CryoEM structures, AlphaFold3-predicted native olfactory receptors, and their water-soluble QTY variants

We display the three types of olfactory receptors, including the CryoEM structures, AlphaFold3-predicted native olfactory receptors, and their water-soluble QTY variants, side by side with the same structural orientations to show their similarities although the native structures are hydrophobic and the QTY variant structures have become hydrophilic (cyan color) (Figure 2).

## Superpositions of CryoEM structures, AlphaFold3-predicted native olfactory receptors, and their water-soluble QTY variants

We superposed the AlphaFold3-predicted native structures of the native olfactory receptors with their respective QTY variants as well

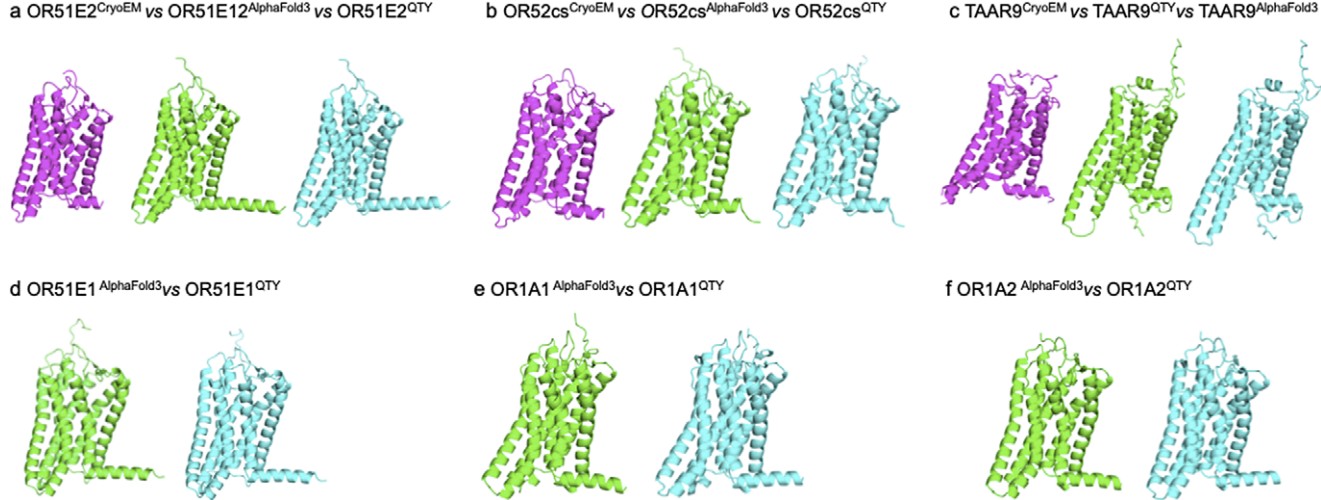

**Figure 2. Side-by-side structural comparison of CryoEM structures (magenta) and the AlphaFold3-predicted native olfactory receptors (green) and their water-soluble QTY variants (cyan)**. These proteins are shown in identical orientations: a) OR51E2$^{CryoEM}$ versus OR51E2$^{AF3}$ versus OR51E2$^{QTY}$, b) OR52cs$^{CryoEM}$ versus OR52cs$^{AF3}$ versus OR52cs$^{QTY}$, c) TAAR9$^{CryoEM}$ versus TAAR9$^{AF3}$ versus TAAR9$^{QTY}$, d) OR51E1$^{AF3}$ versus OR51E1$^{QTY}$, e) OR1A1$^{AF3}$ versus OR1A1$^{QTY}$, f) OR1A2$^{AF3}$ versus OR1A2$^{QTY}$. The flexible C-terminus of OR51E2$^{CryoEM}$ was deleted for structural determination, and thus is a bit shorter compared to the native OR51E2$^{AF3}$ and OR51E2$^{QTY}$ variants.

as their experimentally determined CryoEM structures (only available for OR51E2, OR52cs, and TAAR9). Therefore, the superposition of the following proteins was carried out: OR51E2$^{CryoEM}$ versus OR51E2$^{AF3}$ versus OR51E2$^{QTY}$, OR52cs$^{CryoEM}$ versus OR52cs$^{AF3}$ versus OR52cs$^{QTY}$, TAAR9$^{CryoEM}$ versus TAAR9$^{AF3}$ versus TAAR9$^{QTY}$, OR51E1$^{AF3}$ versus OR51E1$^{QTY}$, OR1A1$^{AF3}$ versus OR1A1$^{QTY}$, and OR1A2$^{AF3}$ versus OR1A2$^{QTY}$ (Figure 3).

For the three olfactory receptors with experimentally determined CryoEM structures (OR51E2, OR52cs, and TAAR9), the native CryoEM structures and their AlphaFold3-predicted QTY variants superpose very well (Figure 3a–c). The root mean square deviation (RMSD) between the native CryoEM structures and the AlphaFold3-predicted QTY variants was between 0.987 Å and 1.275 Å. Therefore, all pairs had an RMSD of <1.30 Å: OR51E2-$^{CryoEM}$ versus OR51E2$^{QTY}$ (1.275 Å), OR52cs$^{CryoEM}$ versus OR52cs$^{QTY}$ (1.038 Å) and TAAR9$^{CryoEM}$ versus TAAR9$^{QTY}$ (0.987 Å). These affirm the significant degree of similarity between the native olfactory receptors and their water-soluble QTY variants, as well as support AlphaFold3's competence and power.

As shown in Figure 3d–f, these structures superposed remarkably well, sharing very similar folds despite a 43.03%–50.31% replacement of amino acids in the transmembrane domain of the QTY variants. Negligible discrepancies and variations are observed. The RMSD values between the AlphaFold3-predicted native structures of the olfactory receptors and their QTY variants are OR51E2$^{AF3}$ versus OR51E2$^{QTY}$ (0.461 Å), OR52cs$^{AF3}$ versus OR52cs$^{QTY}$ (0.441 Å), TAAR9$^{AF3}$ versus TAAR9$^{QTY}$ (0.822 Å), OR51E1$^{AF3}$ versus OR51E1$^{QTY}$ (0.713 Å), OR1A1$^{AF3}$ versus OR1A1$^{QTY}$ (0.713 Å), and OR1A2$^{AF3}$ versus OR1A2$^{QTY}$

(0.732 Å). The range of the RMSD values was between 0.441 and 0.822 Å; thus, all pairs had an RMSD of <1.00 Å. These results not only show the exceptional structural correspondence between native olfactory receptors and their water-soluble QTY variants but also highlight the accuracy of AlphaFold3's predictions.

To further validate the accuracy of AlphaFold3, we also asked how well the CryoEM-determined native olfactory receptor structures would superpose with the AlphaFold3-predicted native structures. The CryoEM and AlphaFold3-predicted native structures superposed very well: OR51E2$^{CryoEM}$ versus OR51E2$^{AF3}$ (1.079 Å), OR52cs$^{CryoEM}$ versus OR52cs$^{AF3}$ (0.832 Å), and TAAR9$^{CryoEM}$ versus TAAR9$^{AF3}$ (1.265 Å). All pairs had an RMSD of <1.30 Å, affirming the advanced performance of AlphaFold3.

### Analysis of the hydrophobic surface of native olfactory receptors and their water-soluble QTY variants

The native olfactory receptors are highly hydrophobic, particularly in the transmembrane alpha-helical domains. This is because the transmembrane alpha helices are embedded directly in the lipid bilayer, and the hydrophobic side chains of amino acids leucine (L), isoleucine (I), valine (V), and phenylalanine (F) interact with the lipid bilayer, thereby excluding water molecules. Consequently, the transmembrane domains demonstrate highly hydrophobic patches (Figure 3). Once these native proteins have been extracted from the lipid bilayer membranes, they require surfactants to solubilize and stabilize. Otherwise, they aggregate and precipitate, losing their biological functions.

After the QTY code was applied, pairwise-replacing the hydrophobic amino acids L, I/V, and F with the hydrophilic amino Q, T,

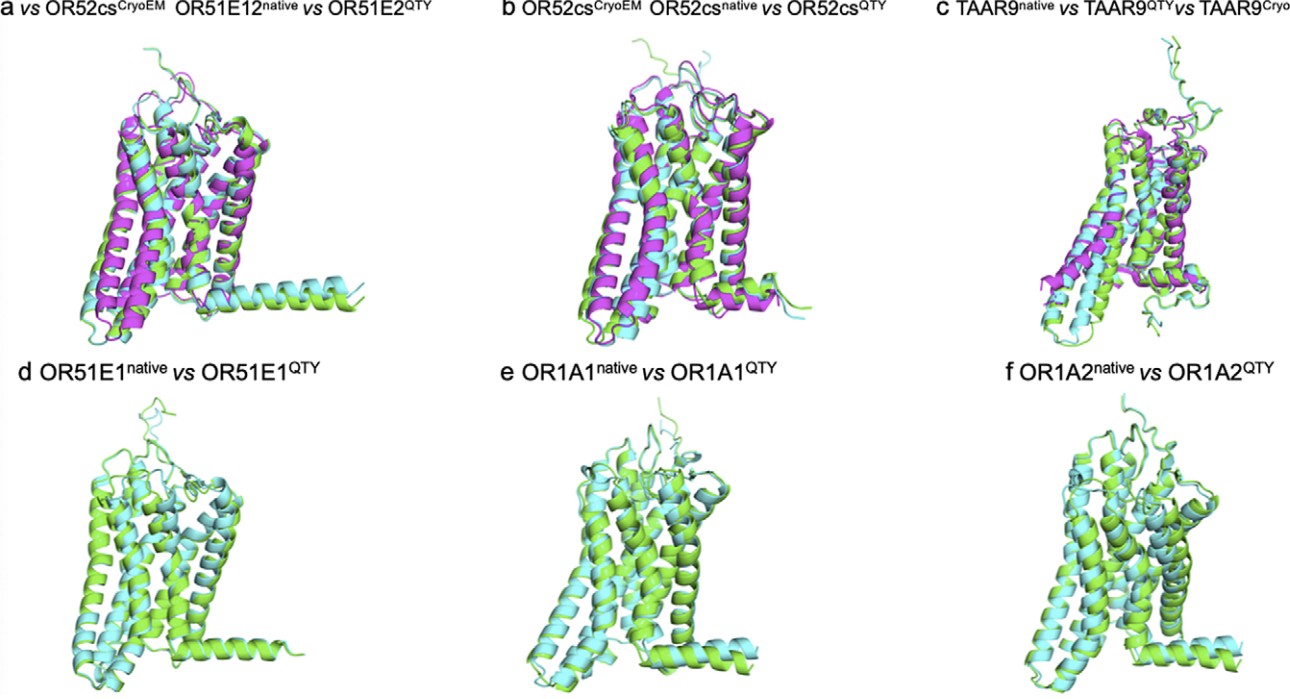

**Figure 3. Superpositions of the AlphaFold3-predicted native structures of six olfactory receptors with their water-soluble QTY variants and CryoEM structures.** Superposition of i) the AlphaFold3-predicted native structures of the olfactory receptors (green) with ii) the AlphaFold3-predicted water-soluble QTY variant structures (cyan) and iii) the experimentally determined CryoEM structures (magenta). These superpositions are shown in Figure 3. These three different kinds of structures are apparently superposed very well. The differences and variations are insignificant. Not only do these three kinds of superpositions further validate the usefulness of AlphaFold3, but they also show that the water-soluble QTY variant olfactory receptors could be used as soluble antigens to generate therapeutic monoclonal antibodies. a) OR51E2$^{CryoEM}$ versus OR51E2$^{AF3}$ versus OR51E2$^{QTY}$, b) OR52cs$^{CryoEM}$ versus OR52cs$^{AF3}$ versus OR52cs$^{QTY}$, c) TAAR9$^{CryoEM}$ versus TAAR9$^{AF3}$ versus TAAR9$^{QTY}$, d) OR51E1$^{AF3}$ versus OR51E1$^{QTY}$, e) OR1A1$^{AF3}$ versus OR1A1$^{QTY}$, f) OR1A2$^{AF3}$ versus OR1A2$^{QTY}$. The flexible C-terminus of OR51E2$^{CryoEM}$ was deleted for structural determination, thus, is a bit shorter compared to the native OR51E2$^{AF3}$ and OR51E2$^{QTY}$ variants.

and Y, the hydrophobic patches were significantly reduced (Figure 4). Transforming the transmembrane alpha-helices from hydrophobic to hydrophilic using the QTY code did not significantly alter their transmembrane structures.

## AlphaFold3 predictions

Over the decades, the scientific community has been attempting to predict the complex process of how proteins naturally fold instantaneously into their 3D structure, one of the most important challenges in the biological sciences. Predicting protein folding is important in advancing our understanding of disease and biological functions. Moreover, it is crucial in drug development and discovery as well as in the creation of biotechnological applications. Numerous attempts have been made to predict how proteins fold; however, this has been an extremely difficult task until the release of AlphaFold in late 2019, which is an AI and machine learning protein structure prediction software.

On May 8, 2024, DeepMind launched AlphaFold3, marking a significant milestone in allowing us to study proteins and their complexes with other molecules. AlphaFold3 goes beyond just the prediction of protein structures and interactions to that of other critical biomolecules including DNA, RNA, other proteins, peptides, and small molecular ligands. AlphaFold3 offers increased accuracy to AlphaFold2 in predicting single protein structures and predicts protein complexes with much higher precision, outperforming previous classical docking tools.

AlphaFold3's predictions allow us to continue to study integral transmembrane protein structures and interactions in silico with increased speed and accuracy. We have used AlphaFold3 to predict the native structure of olfactory receptors as well as their water-soluble QTY variants. Then, we can superpose and compare the native structure of the proteins with the AlphaFold3-predicted QTY variants. Our work using AlphaFold3 has shown that the water-soluble QTY-variant structures are very similar to the native structures, indicating that the QTY code likely works for other transmembrane proteins.

AlphaFold3 offers increased accuracy but also unprecedented speed when it comes to predicting the structure of our native membrane proteins and their QTY variants. Past protein structure predictions could take hours, but AlphaFold3 has the capability to predict complex structures in mere minutes. This efficiency not only accelerates protein structure research and designing new proteins but also facilitates the speeding up of drug discovery and screening.

## Odorant binding and residue-wise analysis

Odorant octanoic acid (OCA) can bind several olfactory receptors with different affinity. The olfactory receptor OR1A2 protein was selected for molecular dynamics studies based on initial virtual screening results from SwissDock, where it exhibited higher docking scores for octanoic acid compared with other candidate olfactory receptors although other receptors also bind to octanoic acid to a lesser extent. The structural analysis (Figure 5a) reveals that octanoic acid is nestled within a hydrophobic pocket of the membrane protein, interacting predominantly with hydrophilic residues LYS109, HIS159 that form hydrogen bonds, and hydrophobic residues ILE181 and PHE206 that form hydrophobic interactions. For protein design, these key residues might need to be preserved to maintain function. The electrostatic potential surface demonstrates the compatibility of the binding pocket for the hydrophobic tail and polar head group of octanoic acid, suggesting a stable interaction within this region.

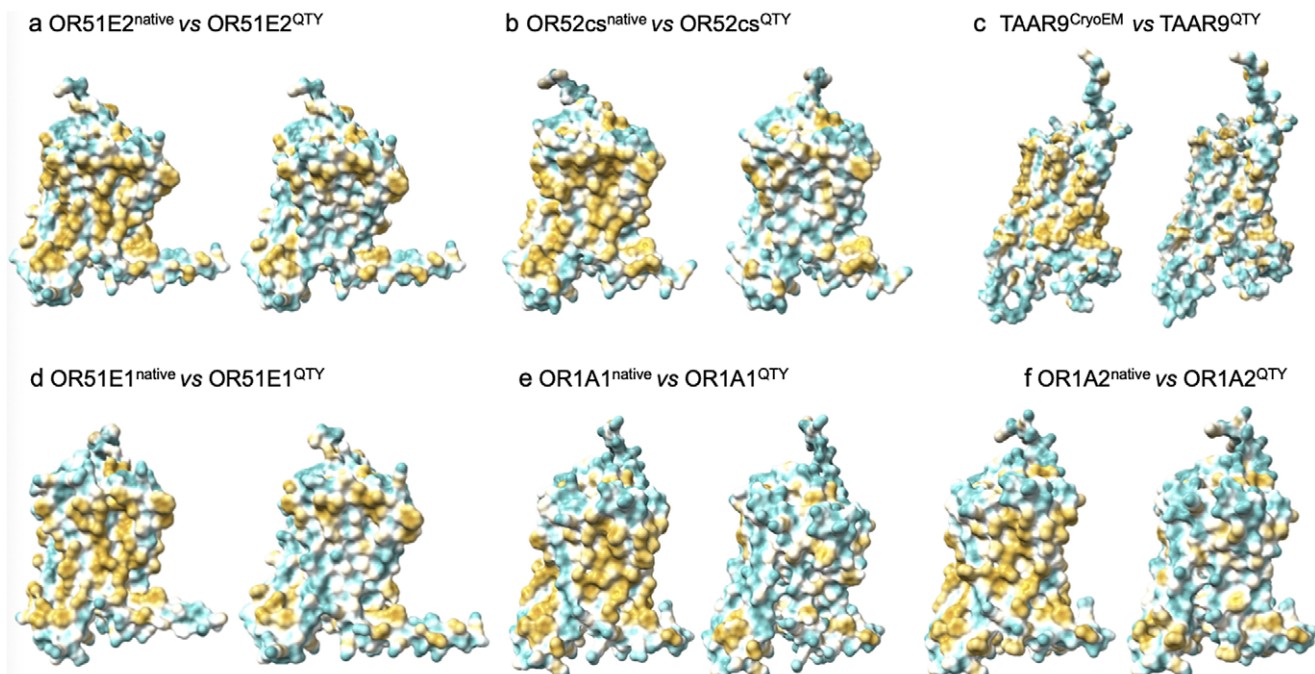

**Figure 4. The hydrophobic surface of six native olfactory receptor proteins and their water-soluble QTY variants**. The native olfactory receptors have many hydrophobic residues L, I, V, and F in the transmembrane helices. After Q, T, and Y substitutions of L, I and V, and F, respectively, the hydrophobic surface patches (yellowish) in the transmembrane helices become more hydrophilic (cyan). a) OR51E2$^{CryoEM}$ versus OR51E2$^{AF3}$ versus OR51E2$^{QTY}$, b) OR52cs$^{CryoEM}$ versus OR52cs$^{AF3}$ versus OR52cs$^{QTY}$, c) TAAR9$^{CryoEM}$ versus TAAR9$^{AF3}$ versus TAAR9$^{QTY}$, d) OR51E1$^{AF3}$ versus OR51E1$^{QTY}$, e) OR1A1$^{AF3}$ versus OR1A1$^{QTY}$, f) OR1A2$^{AF3}$ versus OR1A2$^{QTY}$. The flexible C-terminus of OR51E2$^{CryoEM}$ was deleted for structural determination, thus, is a bit shorter compared to the native OR51E2$^{AF3}$ and OR51E2$^{QTY}$ variants.

The 50 ns molecular dynamics simulations in the membrane system and subsequent Molecular Mechanics Poisson-Boltzmann Surface Area (MMPBSA) binding free energy calculations revealed insights into the binding interactions between the odorants and the OR1A2. The total binding energy of octanoic acid to the protein fluctuated around an average of −13 kcal/mol, indicating a moderate to strong binding affinity (Figure 5b). The radius of gyration was also maintained through the simulation which was compared with the OR1A2 membrane system without ligands. The moving average trendline suggests stabilization of the binding interaction over the

course of the simulation, with minor fluctuations indicating transient conformational changes in the protein–ligand complex. Among the OR1A2 residues, LYS109 demonstrated the most significant contribution to binding, with a median energy of −54.435 kcal/mol (Table 2, Figure 5c). The narrow interquartile range (IQR), −55.775 to −52.7725 kcal/mol, for LYS109, indicates consistency in its binding contribution throughout the simulation. Other important contributors include HIS159 (in its protonated HSD form), ILE 181, and PHE 206. In substitution-based protein designs, such as the QTY code, maintaining consistent contributions from key residues

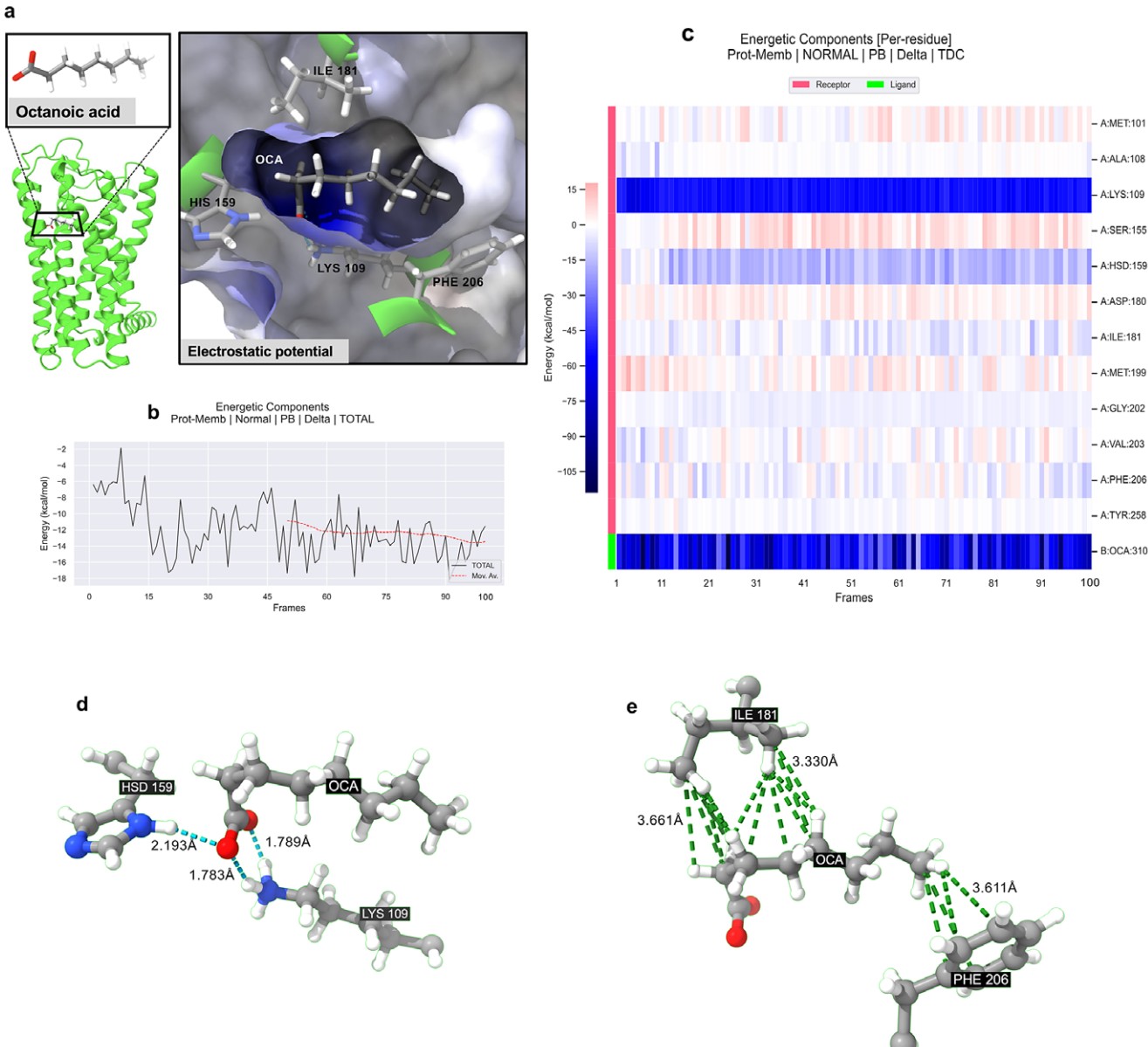

**Figure 5. Binding interactions and energetic analysis of octanoic acid-OR1A2 complex.** a) Structural representation of the OR1A2 complex bound with octanoic acid. The left panel shows the protein structure in green cartoon representation, with the ligand octanoic acid shown as a stick model. The inset magnifies the binding site, highlighting key interacting residues – HIS159, LYS109, ILE181, and PHE206 – along with the electrostatic potential surface of the binding pocket. Octanoic acid is positioned within a hydrophobic cleft, interacting predominantly with LYS109. b) MMPBSA calculated the total binding energy of octanoic acid to the membrane protein over 100 simulation frames (50 ns). The black line represents the binding energy, while the red dashed line indicates the moving average trend across the frames. The graph shows fluctuations in the energy, stabilizing at an approximate total binding energy of around −13 kcal/mol. c) Heatmap of per-residue energetic contributions to octanoic acid binding (B:OCA) across 100 simulation frames (50 ns). The heatmap shows the contributions of individual residues (on the y-axis) to the binding energy (indicated by the color scale, red to blue). Significant negative energy contributions are observed for receptor (A) residues LYS 109 and HSD 159. HSD = Histidine with a protonated delta nitrogen (ND1). d) shows the hydrogen bindings of octanoic acid from HIS159 and LYS109. The histidine side chain acts as a hydrogen donor (2.193 Å) to one of the two oxygens on octanoic acid. The lysine side chain acts as two hydrogen donors (1.783 and 1.789 Å) to both oxygens on octanoic acid. e) shows the hydrophobic interactions between ILE181 (3.330 and 3.661 Å) and the PHE206 (3.611 Å) to the carbon chain of octanoic acid.

**Table 2.** Residue-wise Decomposition Analysis of Molecular Mechanics Poisson–Boltzmann Surface Area (MMPBSA) binding free energy calculations for OR1A2-octanoic acid complex

| Residue[a] | Median binding free energy (kcal/mol) | IQR1 (lower quartile)[b] | IQR3 (upper quartile)[c] |
|---|---|---|---|
| L:OCA (octanic acid) | −61.825 | −78.4725 | −49.8475 |
| R:LYS:109 | −54.435 | −55.775 | −52.7725 |
| R:HIS:159 | −16.02 | −18.52 | −10.7325 |
| R:GLY:202 | −3.67 | −4.2825 | −2.8025 |
| R:ILE:181 | −3.46 | −5.905 | −1.23 |
| R:PHE:206 | −2.195 | −6.385 | 0.9575 |
| R:TYR:258 | −1.535 | −2.515 | −0.6525 |
| R:ALA:108 | −1 | −1.7225 | 0.0275 |
| R:VAL:203 | −0.925 | −4.2025 | 1.83 |
| R:MET:101 | 0.02 | −2.0125 | 4.1925 |
| R:MET:199 | 2.44 | −1.7 | 6.0875 |
| R:ASP:180 | 3.1 | 0.92 | 5.345 |
| R:SER:155 | 5.435 | 1.5775 | 8.5825 |

[a]Residue contributions for octanoic acid binding (L:OCA) across 100 simulation frames (50 ns). R = receptor, OR1A2. L: ligand, octanoic acid.
[b]The lower quartile corresponds with the 25 percentile.
[c]The upper quartile corresponds with the 75 percentile.

might be necessary to ensure that the protein remains functional after substitutions.

Figure 5d shows the hydrogen bindings of octanoic acid from HIS159 and LYS109. The histidine side chain acts as a hydrogen donor (2.193 Å) to one of the two oxygens on octanoic acid. The lysine side chain acts as two hydrogen donors (1.783 Å and 1.789 Å) to both oxygens on octanoic acid. Figure 5e shows the hydrophobic interactions between ILE181 (3.330 and 3.661 Å) and PHE206 (3.611 Å) to the carbon chain of octanoic acid.

Interestingly, some residues showed wider IQRs, suggesting a degree of flexibility in the OR1A2 binding pocket. For instance, residues PHE206, VAL203, and MET101 show particularly wide IQRs, spanning both negative and positive energy contributions. This flexibility could be indicative of the receptor's potential to accommodate other odorants. This is particularly relevant for designing variants that retain the ability to bind multiple ligands. The binding interactions of octanoate, the deprotonated form of octanoic acid, with the OR1A2, were analyzed to understand the similarities and differences in their binding patterns. Despite the difference in their protonation states, the overall binding orientation and key residue interactions remain consistent, suggesting that the binding site is capable of stabilizing both forms of the ligand. Octanoate displayed a similar energy profile to octanoic acid, indicating that the deprotonation of the carboxyl group does not significantly alter the binding strength. Furthermore, the moving average trendlines for both ligands suggest that their binding interactions are stable over time (Figure 5b).

### An amine odorant spermidine-TAAR9 binding structural bioinformatic study

The odorant binding pattern extends beyond the common olfactory receptor family, as evidenced by the binding of an amine odorant spermidine binding in trace amine receptor TAAR9, which utilizes a topologically similar pocket (Figure 6a). Molecular Mechanics Poisson–Boltzmann Surface Area (MMPBSA) binding free energy calculations revealed both systems were energetically favorable (Figure 6b). This cross-receptor spatial conservation of binding sites suggests a potentially convergent evolutionary mechanism in odorant sensing G protein-coupled receptors. Interestingly, comparative assessment of the hydrogen bonding networks demonstrated notable differences between the two ligands, despite their spatial overlap within the binding pocket. The predominant positive electrostatic potential (red coloration) in the spermidine pocket of TAAR9 suggests a binding site optimized for interaction with negatively charged or electronegative ligands, consistent with its role in recognizing biogenic amines (Figure 6a).

Conversely, the negative electrostatic potential (blue coloration) characterizing the octanoic acid binding pocket in OR1A2 indicates a binding potential for electropositive or partially positive charged regions of odorant molecules (Figure 5a). This electrostatic complementarity between receptor-binding pockets and their respective ligands indicates the molecular basis for selective olfactory recognition, where the spatial distribution of charge plays a crucial role in determining receptor–ligand specificity. Concurrently, interacting residues were also diverse, as ASP 112 showed a significant contribution to binding energy for TAAR9 (Figure 6c). Protonated histidine residues showed contributions for both ligands (Figure 6d,e).

### Future scopes and the potential applications

The consistent binding trends observed across our simulations provide a computational foundation for odorant interactions and could guide future experimental investigations. While we identified topologically similar binding pockets in these receptors, the specific molecular interactions driving ligand recognition were found to be distinct, further studies could involve pocket flexibility and conformational changes. Especially considering the prediction of the binding pocket flexibility of these receptors is a daunting task (Wang et al., 2024). Lipid distortion profiling could produce additional insights into the membrane-specific impact on protein interactions (Karagöl et al., 2024c), including odorant binding and pocket flexibility. Further molecular dynamics simulations on QTY-variants and a comparative analysis could also be helpful in understanding structural flexibility in odorant binding. Recent simulations on QTY-variant glutamate transporters revealed lipid impact and structural flexibility in the transporter architecture (Karagöl et al., 2024c). Regardless, the stable binding energies observed during simulations indicate physiologically relevant interactions, opening possibilities for medical applications in treating olfactory disorders and developing new drug delivery systems targeting membrane proteins. Olfactory receptors are also distributed in non-olfactory tissues, making them promising targets for a range of diseases (Wu et al., 2024).

These findings could guide the understanding of the evolutionary relationships between olfactory and trace amine-associated receptors and develop broader computational frameworks for predicting ligand specificity across related receptor families. Our results highlight the significant role of electrostatic potential in ligand interactions within these binding pockets. Notably, QTY-variants, which preserve residue charge, hold the potential for designing artificial receptors. The detailed binding characteristics uncovered in this study may also support advancements in artificial

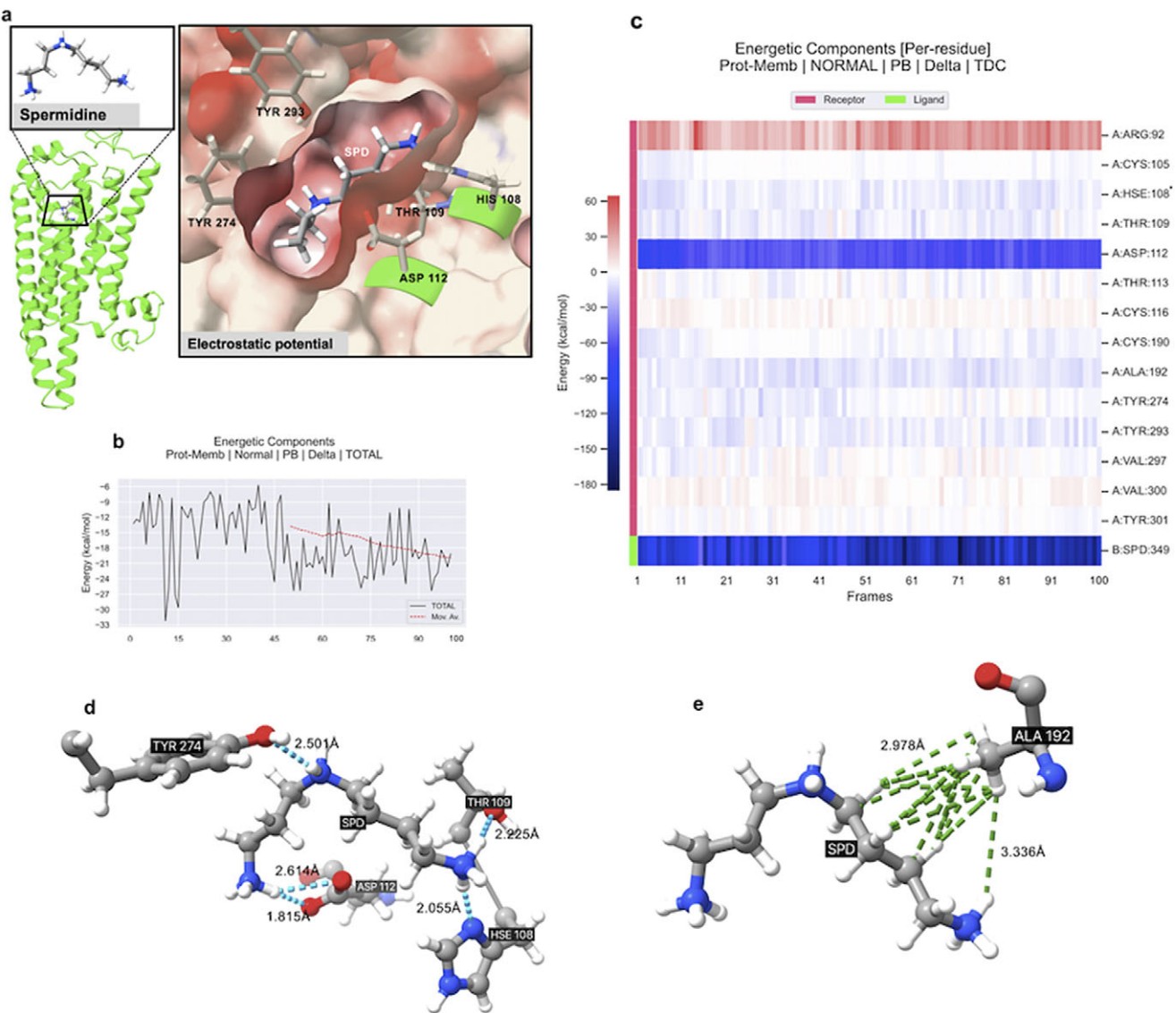

**Figure 6. Binding interactions and energetic analysis of spermidine–TAAR9 complex.** a) Structural representation of the TAAR9 complex bound with spermidine (SPD). The left panel shows the protein structure in green cartoon representation, with the ligand shown as a stick model. The inset magnifies the binding site, highlighting key interacting residues – TYR274, TYR293, HIS108, THR109, and ASP112 – along with the electrostatic potential surface of the binding pocket. b) MMPBSA calculated the total binding energy of the complex in the membrane over 100 simulation frames (50 ns). The black line represents the binding energy, while the red dashed line indicates the moving average trend across the frames. The graph shows fluctuations in the energy, stabilizing at an approximate total binding energy of around −18 kcal/mol. c) Heatmap of per-residue energetic contributions to spermidine binding (B:SPD) across 100 simulation frames (50 ns). The heatmap shows the contributions of individual residues (on the $y$-axis) to the binding energy (indicated by the color scale, red to blue). Significant negative energy contributions are observed for receptor (A) residues ASP 112. HSE = Histidine in epsilon-protonated state. d) shows the hydrogen bindings of spermidine from TYR274, HIS108, THR109, and ASP112. e) shows the hydrophobic interactions between ALA192 (2.978 and 3.336 Å) to the carbon chain of spermidine.

olfactory sensor development. To this extent, odorant receptor structure construction could be used to identify potential modulators (Wang et al., 2024). Alongside the olfactory system, these findings could provide insights into developing new delivery systems for drugs targeting membrane proteins. The observed plasticity in hydrogen bonding patterns may also inform structure-based drug design strategies for other G protein-coupled receptors where similar mechanical principles might apply.

## Conclusion

In our study, we selected six human olfactory receptors including OR51E2, OR52cs, TAAR9, OR51E1, OR1A1, and OR1A2 that have

been linked with research for the possible development of cancer treatment and detection methods. We applied the QTY code to the six olfactory proteins to convert the hydrophobic alpha helices to hydrophilic alpha helices and thus create water-soluble QTY variants. Then, we used AlphaFold3 to predict the structures of both the native proteins and their QTY variants. Through superposing the QTY variants and the native structures and calculating the RMSD values, we found that despite the substantial replacement of the amino acids in the transmembrane domains, the structures of the QTY variants were remarkably similar to those of the native proteins. This reveals that the AlphaFold3-predicted QTY variants of the native olfactory receptors are likely to retain their properties. Using bioinformatic computational tools, we validated this by analyzing calculated characteristics affiliated with protein stability

and water solubility. Finally, we found that the surfaces of the QTY variants were markedly more hydrophilic than those of the native proteins. We also carried out the odorant binding molecular simulation study. Such a study revealed that the odorant octanoic acid forms a complex with the olfactory receptor OR1A2 with specific amino acid interactions. These water-soluble olfactory receptor QTY variants may now be engineered to design and develop as biomimetic sensing devices (Qing et al., 2023). Our current studies further demonstrate that the QTY code is a valid method to accurately design water-soluble variants of olfactory receptors. We believe these hydrophilic QTY variants of the olfactory receptors have potential for cancer detection technology and the discovery of therapeutic treatments for various cancerous diseases.

## Methods

### Protein sequence alignments and other characteristics

The native protein sequences for OR51E2, TAAR9, OR51E2, OR1A1, and OR1A2 were attained from UniProt (https://www.uniprot.org). Professor Hee-Jung Choi of Seoul National University, Korea, kindly provided us with the native protein sequence for OR52cs after we could not find it on UniProt or in any citation references. The native protein sequences and those of their QTY variants were aligned using the methods previously described. We used the Expasy website (https://web.expasy.org/compute_pi/) to calculate the MWs and pI values of the proteins.

### AlphaFold3 predictions

We predicted the structures of the native proteins and their QTY variants using the AlphaFold3 website (https://alphafoldserver.com), following the included instructions. We also used the UniProt website (https://www.uniprot.org) to obtain protein ID, entry name, description, and FASTA sequence for each native protein. We then applied the QTY code to the FASTA sequences by manually replacing amino acids in the transmembrane domains found on the UniProt website and confirmed using the Protter website (https://wlab.ethz.ch/protter/start/).

### Superposed structures

PDB files for native protein structures determined experimentally using Cryogenic electron microscopy (Cryo-EM) were taken from the RCSB PDB, including OR51E2, OR52cs, and TAAR9. We used AlphaFold3 (https://alphafoldserver.com) to predict the native structures of the olfactory receptors as well as their QTY variants. PyMol (https://pymol.org) was used to superpose these structures and calculate their RMSD values. Predictions for the QTY variants were carried out using AlphaFold3 (https://alphafoldserver.com). Then, these structures were superposed, and the RMSD values were calculated using PyMOL (https://pymol.org). For OR51E2, TAAR9, and OR52cs, the CryoEM molecular only model the heterodimer. As the AlphaFold2-predicted QTY variants only model the monomer, we removed unstructured loops and other protein monomers from the figures for clarity.

### Structure visualization

We first used PyMOL (https://pymol.org) to superpose the native predicted protein structures, their QTY variants, and the CryoEM-determined structures for those proteins where these existed. We

then used UCSF Chimera (https://pymol.org) to render each protein model with hydrophobicity patches.

### Odorant docking and molecular dynamics simulations

For the docking studies, octanoic acid is selected as a ligand, represented using SMILES notation. AlphaFold2 database predicted proteins were utilized as receptors. Molecular docking was performed using the Attracting Cavities 2.0 (AC) method available on the SwissDock server (http://www.swissdock.ch/) (Röhrig et al., 2023; Bugnon et al., 2024). The OR1A2 was chosen for molecular dynamics studies as the target receptor because of its higher scores in SwissParam assessments. Molecular dynamics simulations were conducted for the membrane systems of OR1A2, OR1A2-octanoic acid complex, and OR1A2-octanoate complex with the best results from the molecular docking study according to SwissParam scores. For TAA9, spermidine was selected as a ligand as it is an identified agonist (Xu and Li, 2020). The internal binding pocket was predicted using the SwissDock server, based on SwissParam scoring. All MD simulations and analyses were executed on Google Colab (https://colab.research.google.com), via Ubuntu 2021 4.2, utilizing a total of 96 core v2 TPUs and 334GB RAM. The simulations were parallelized across multiple processors or cores within the VM. Configuration files and Linux bash codes for the simulations are publicly available with step-by-step instructions.

Membrane-protein systems were constructed using CHARMM-GUI's web-based membrane builder (Jo et al., 2008; Wu et al., 2014). The protein portion was centered in a rectangular box and protonation states were assigned based on the local pH. The spatial orientation relative to the lipid bilayer was optimized via the PPM 2.0 method (Lomize et al., 2012). This method accounts for the anisotropic water-lipid environment, characterized by dielectric constant and hydrogen-bonding profiles (Lomize et al., 2012). The membrane models generated consisted of 70% 1-palmitoyl-2-oleoyl-glycero-3-phosphocholine (POPC) and 30% cholesterol, as it represents a simplified model of the plasma membrane. The system was solvated in TIP3P water with 150 mM KCl. All molecular dynamics (MD) simulations were conducted using GROMACS 2021.4 (Abraham et al., 2015) with the CHARMM36m all-atom force field (Huang et al., 2017). System energy was minimized using the steepest descent until maximum forces converged below 1000 kJ/mol/nm. Electrostatics were handled with Particle Mesh Ewald (PME), with both Coulomb and van der Waals interaction cutoffs set at 1.2 nm. A multistep minimization and equilibration process was employed to relax the protein-membrane systems. The standard six-step CHARMM-GUI protocol (Jo et al., 2008) was used for 125-ps equilibration simulations. Temperature and pressure were maintained at 303.15 K and 1 bar, respectively, using the Nose–Hoover thermostat and Parrinello–Rahman barostat with semi-isotropic coupling. Following NVT and NPT equilibration, a 50-ns production MD simulation was run, with timestamps every 500 ps. Trajectories were subsequently combined using gmx traj (Abraham et al., 2015).

System stability was assessed through trajectory analysis. This included calculating the protein gyration radius (gmx gyrate tool), and residue root mean square fluctuation (RMSF) for protein Cα atoms. The solvent-accessible surface area (SASA) of protein residue side chains was determined using gmx_sasa, with a solvent probe radius of 1.4 Å (Huang et al., 2017). The resulting plots were rendered using Grace (https://plasma-gate.weizmann.ac.il/Grace/). MMPBSA binding free energy estimations were performed on the full 50 ns of equilibrated MD trajectories.

## MMPBSA calculations for membrane-protein systems

The binding free energy of the complexes was calculated using the Molecular Mechanics Poisson-Boltzmann Surface Area (MMPBSA) algorithm and performed using the gmx_MMPBSA tool, following the Amber reference manual for membrane-bound protein systems (Miller et al., 2012; Valdés-Tresanco et al., 2021). Implicit membrane region incorporated into the solvation calculations. With the default options, the program computed solvent-excluded surfaces using both the water probe (prbrad = 1.40) and the membrane probe (mprob = 2.70). Electrostatic energy and forces were computed using the particle-particle particle-mesh (P3M) method (Botello-Smith and Luo, 2015). Binding energies were calculated for each time step, with averages and standard deviations reported. The standard error of the mean (SEM) was determined using error propagation. Per-residue decomposition analysis (idecomp = 2) was conducted to assess individual residue contributions to binding energy. This included 1–4 EEL and 1–4 VDW terms in total EEL and VDW potentials, respectively. Residues within 4 Å of both receptor and ligand were included in the output. Binding surface compositions were further analyzed and visualized using the gmx_MMPBSA_ana tool.

**Open peer review.** To view the open peer review materials for this article, please visit http://doi.org/10.1017/qrd.2024.18.

**Supplementary material.** The supplementary material for this article can be found at http://doi.org/10.1017/qrd.2024.18.

**Data availability statement.** The AlphaFold2-predicted protein structures are at European Bioinformatics Institute (EBI) https://alphafold.ebi.ac.uk. The QTY code-designed water-soluble variants are published in this article and at Protein characteristics used in the analysis are available on UniProt, https://www.uniprot.org/. The native cryo-EM-determined ABC-transporter proteins are available in the RCSB PDB repository, https://www.rcsb.org/. The QTY code-designed water-soluble variants of the proteins are available at https://github.com/karagol-taner/Olfactory-receptors-QTY.

**Acknowledgements.** We thank Prof. Hee-Jung Choi of the Department of Biological Sciences, Seoul National University, Republic of Korea for kindly providing us with the protein sequence of consensus OR52cs by email. Prof. Hee-Jung Choi's lab determined the CryoEM structure of OR52cs in December 2023.

**Author contribution.** Conceptualization: S.Z.; Formal analysis: F.J., A.K., T.K.; Investigation: Methodology: F.J., A.K., T.K.; Validation: F.J., A.K., T.K.; Data curation: A.K., T.K.; Writing—original draft preparation: F.J., A.K., T.K, S.Z.; Review and editing: F.J., A.K., T.K. and S.Z.

**Financial support.** Finn Johnsson is a high school student in London, UK. Taner Karagöl and Alper Karagöl are medical school students. There is no financial support for this digital structural bioinformatic study only free online tools.

**Competing interest.** Massachusetts Institute of Technology (MIT) filed several patent applications for the QTY code for GPCRs excluding the olfactory receptors. OH$_2$Laboratories licensed the technology from MIT to work on water-soluble GPCR variants. S.Z. is an inventor of the QTY code and has a minor equity in OH$_2$Laboratories. S.Z. is a Scientific Advisor and has minor shares for a startup RealNose to develop a sensing device based on olfactory receptors. S.Z. founded a startup 511 Therapeutics to generate therapeutic monoclonal antibodies against solute carrier transporters to treat pancreatic cancer. S.Z. has majority equity in 511 Therapeutics. All other authors have no competing interests.

**Additional statement.** 1) All methods were carried out in accordance with relevant guidelines and regulations. 2) All experimental protocols were approved by a named institutional and licensing committee. 3) Neither human biological samples nor human subjects were used in the study. This is a completely digital structural bioinformatic study using the publicly available AlphaFold3 machine learning program.

**Ethics Statement.** All methods were carried out in accordance with relevant guidelines and regulations. All experimental protocols were approved by a named institutional and licensing committee. Neither human biological samples nor human subjects were used in the study. This is a completely digital structural bioinformatic study using the publicly available AlphaFold3 machine learning program.

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
