## [Reviewer Report]

This manuscript studies structures and properties of olfactory receptors and their designed water-soluble variants using bioinformatic tools, such as AlphaFold3 and SwissDock. The author took a step forward from his previous reports on QTY designed membrane proteins to apply on six olfactory receptors. The binding between OR1A2 receptor and octanoate was analyzed through AlphaFold3 and molecular dynamic simulations, providing residue-wise bioinformatics analysis. The study is well-presented and is suitable for publication in QRB Discovery following minor revision.

1. Please check the formatting of the manuscript and ensure consistency. For instance, the penultimate reference on page 11 provides DOI but not for other references.

2. The manuscript could benefit from more discussion on the limitations and difficulties of current olfactory receptor studies in introduction.

3. The manuscript could benefit from more discussion on the molecular simulation of odorant binding. What are the potential significance for the treatment of related diseases and signal transduction mechanisms?

---

## [Reviewer Report]

Comments: Johnsson et al. manuscript “Structural bioinformatic study of six human

olfactory receptors and their AlphaFold3 predicted water-soluble QTY variants and

OR1A2 with an odorant octanoate”

This paper is an interesting original study on six human olfactory receptors where the

QTY code was applied to transform the membrane-based olfactory receptors into

water-soluble variants. The authors made the striking observation that when

superimposing the native receptors with the QTY variants, a strong resemblance and

similarity between the original receptor structure and QTY variants were found. This

is a particularly remarkable finding since profound structural changes were

introduced to convert the hydrophobic alpha-helix domains of the trans-membrane

region of the odorant receptors to hydrophilic alpha-helices. In addition, a ligandbinding

molecular simulation study was executed with the odorant octanoate and the

water-soluble olfactory receptor QTY variant of OR1 A2. Interestingly, the octanoate

ligand recognized the QTY variant and formed a complex.

The paper is suitable for publication but should address some minor points:

1. The abstract mentions “olfactory receptors to recognize seemingly unlimited

scents”. It would be advisable to avoid descriptions such as “unlimited” which

are impossible to prove.

2. The abstract mentions further that odorant receptors are “fully embedded in

the cell membrane”. There are important functional parts of the receptor

outside of the membrane. This should be corrected in the text.

3. Two very recent references should be added since they are recent relevant

publications on the subject:

a) A recent review on olfactory receptors by Wang et al., discusses “Modeling

of Olfactory Receptors”, Chapt. 10., in “Homology Modeling”, Springer,

b) Wu et al. “Structure and function of olfactory receptors”, Trends in

Pharmaceutical Studies, 2024